# Methods of Visualizing the Results of an Artificial-Intelligence-Based Computer-Aided Detection System for Chest Radiographs: Effect on the Diagnostic Performance of Radiologists

**DOI:** 10.3390/diagnostics13061089

**Published:** 2023-03-13

**Authors:** Sungho Hong, Eui Jin Hwang, Soojin Kim, Jiyoung Song, Taehee Lee, Gyeong Deok Jo, Yelim Choi, Chang Min Park, Jin Mo Goo

**Affiliations:** 1Department of Radiology, Seoul National University Hospital, Seoul 03082, Republic of Korea; 2Department of Radiology, Seoul National University College of Medicine, Seoul 03082, Republic of Korea; 3Institute of Radiation Medicine, Seoul National University Medical Research Center, Seoul 03082, Republic of Korea

**Keywords:** chest radiography, artificial intelligence, deep learning, computer-aided detection, diagnostic accuracy

## Abstract

It is unclear whether the visualization methods for artificial-intelligence-based computer-aided detection (AI-CAD) of chest radiographs influence the accuracy of readers’ interpretation. We aimed to evaluate the accuracy of radiologists’ interpretations of chest radiographs using different visualization methods for the same AI-CAD. Initial chest radiographs of patients with acute respiratory symptoms were retrospectively collected. A commercialized AI-CAD using three different methods of visualizing was applied: (a) closed-line method, (b) heat map method, and (c) combined method. A reader test was conducted with five trainee radiologists over three interpretation sessions. In each session, the chest radiographs were interpreted using AI-CAD with one of the three visualization methods in random order. Examination-level sensitivity and accuracy, and lesion-level detection rates for clinically significant abnormalities were evaluated for the three visualization methods. The sensitivity (*p* = 0.007) and accuracy (*p* = 0.037) of the combined method are significantly higher than that of the closed-line method. Detection rates using the heat map method (*p* = 0.043) and the combined method (*p* = 0.004) are significantly higher than those using the closed-line method. The methods for visualizing AI-CAD results for chest radiographs influenced the performance of radiologists’ interpretations. Combining the closed-line and heat map methods for visualizing AI-CAD results led to the highest sensitivity and accuracy of radiologists.

## 1. Introduction

Chest radiography is at the forefront of the recent trend of applying artificial intelligence (AI) technology in daily clinical practice. Indeed, AI-based software that can identify various types of abnormalities has been developed and utilized in clinical practice [1,2,3]. Among the various clinical applications, the use of artificial-intelligence-based software as a computer-aided detection (CAD) tool to help radiologists or physicians identify subtle abnormalities has been most widely accepted [1,2,4,5,6].

For artificial-intelligence-based computer-aided detection (AI-CAD) tools, the primary aim is to enhance the detection performance of interpreting radiologists or physicians [2,4,5,6,7]. Therefore, in addition to intrinsic performance, the method of delivering the results of the analysis to physicians is the key component of an AI-CAD tool to demonstrate its efficacy and value in clinical practice. Typically, an AI-CAD tool provides its results with annotations highlighting the location of the detected abnormality overlaid on the input image, as well as a confidence score for the detection [8,9]. Color-coded heat maps and closed lines along the boundary of the abnormality are two representative methods for visualizing AI-CAD results [3,10,11].

Choosing one of the two methods of visualization, or using them in combination, may have an influence on the interaction between the AI-CAD and interpreting physician and the performance of the interpreting physician. However, to date, most studies on the development and validation of AI-CAD focused on the performance of the AI itself rather than on the method of visualization of the result. Therefore, it remains unclear the method of visualization that is optimal for improving the performance of radiologists’ interpretations.

Therefore, the purpose of our study was to investigate the accuracy of radiologists’ interpretation in identifying clinically relevant abnormalities on chest radiographs using AI-CAD with different visualization methods, and to explore the optimal visualization method for AI-CAD for chest radiographs.

## 2. Materials and Methods

### 2.1. Patients

We retrospectively included patients who met the following inclusion criteria: (a) visited the emergency department of a tertiary referral hospital in South Korea between 1 January and 30 June 2017; (b) underwent chest radiography for evaluation of acute respiratory symptoms; and (c) underwent chest CT during their stay in the emergency department.

A total of 249 chest radiographs were obtained from 249 patients (male-to-female ratio, 148:101; mean age ± standard deviation, 62 ± 17 years). Table 1 shows the patients’ demographic and clinical information. A total of 189 (75.9%) chest radiographs were obtained by using a fixed radiography scanner. The most common chief complaint of visiting the emergency department was dyspnea (18.9%), followed by chest pain (15.3%) and fever (11.6%).

In the present study, 49.8% (124/249) of patients were reported in previous studies [12,13]. However, the purpose of previous studies was to evaluate the performance of an AI-CAD to identify clinically significant abnormalities among chest radiographs from patients in the emergency department [12] and to evaluate the calibration of the AI-CAD [13], which was entirely different from that of the present study.

### 2.2. Chest Radiographs

The study included only initial chest radiographs obtained in the emergency department (one radiograph per patient). In cases of multiple visits to the emergency department during the study period, chest radiographs obtained at the initial visit were included.

Posteroanterior and anteroposterior radiographs were included in this study. Posteroanterior radiographs were obtained in an erect position using a single fixed radiography unit (Multix FD; Siemens Helthineers, Erlangen, Germany), while anteroposterior radiographs were obtained in the supine position using a portable radiography scanner (DRX-Revolution; Carestream Health, Rochester, NY, USA).

### 2.3. AI-CAD

Commercially available AI-CAD (Lunit INSIGHT for CXR, version 2.0.3.0; Lunit, Seoul, Korea) was retrospectively applied to chest radiographs. AI-CAD was designed to identify pulmonary nodules, pulmonary infiltration, and pneumothorax on a single frontal chest radiograph, with a confidence score (0–100%) for the presence of identified abnormality [14]. Three different methods were utilized for the visualization of the AI-CAD analysis results (Figure 1 and Figure 2).

Heat map method: A heat map was overlaid on the identified abnormalities. The color of the heat map represents the confidence score. Higher confidence scores are coded in red, while lower confidence scores are coded in blue.

Closed-line method: A closed line without any color information is displayed along the boundary of the identified abnormality. Confidence scores were directly visualized with numbers next to the closed curve.

Combined method: Information visualized using both the heat map and closed-line methods was visualized on a single image.

The threshold confidence score for visualization was 15%, the default value provided by the manufacturer.

### 2.4. Reader Test

Five trainee radiologists (S.K., J.S., T.L., G.D.J., and Y.C.; 1st to 3rd year of residency training) participated in the reader test. The reader test consisted of three interpretations. In each interpretation session, readers interpreted the chest radiographs with AI-CAD results using one of the three visualization methods. Readers were informed that all chest radiographs were obtained from patients who visited an emergency department with acute respiratory symptoms, and the chief complaint of visiting the emergency department was also provided. First, the readers were asked to answer whether there were any clinically significant abnormalities requiring further evaluation or treatment. In case of an abnormality, readers were asked to describe up to three abnormal findings.

In order to minimize the bias caused by repeated interpretation of a single chest radiograph, the sequence of the utilization of each visualization method was randomized for each reader and the chest radiograph. Further, the sequences of chest radiographs in each interpretation session were randomly reshuffled in each interpretation session for all readers (Figure 1). Finally, a wash-out period of at least one month was set up between each interpretation session.

### 2.5. Reference Standard and Performance Metrics

To define the reference standard for the presence of any clinically significant abnormality, a single thoracic radiologist (E.J.H.; 11 years of experience in the interpretation of chest radiographs and chest CTs) reviewed the chest radiographs and the corresponding chest CTs. With reference to chest CT, up to two key abnormal findings that may be associated with patients’ respiratory symptoms and require further evaluation or treatment were defined as the reference standards. Key abnormal findings were classified into six categories: (a) pulmonary nodule or mass, (b) pulmonary air-space opacity, (c) pulmonary interstitial opacity, (d) pleural effusion, (e) pneumothorax, and (f) others. Subtle abnormalities that could not be identified on chest radiographs, even in the retrospective review of chest CT, were excluded from the reference standard.

Sensitivity, specificity, and accuracy were used to evaluate examination-level classification performance. For evaluation of sensitivity and accuracy, the interpretation of readers or AI-CAD results was regarded as true-positive only when at least one key abnormal finding was correctly identified. Sensitivity was defined as the proportion of true-positive interpretations among the radiographs with positive reference standards, and accuracy was defined as the proportion of true-positive and true-negative interpretations among all radiographs.

For evaluation of the lesion-level detection performance, the detection rate (the proportion of correctly identified abnormalities among all clinically significant abnormalities by the reference standard) was used. The detection rate of each abnormality type was also investigated.

### 2.6. Preference Survey

After completing all three sessions of the reader test, readers were asked to complete a questionnaire to survey their subjective preferences for each visualization method. The questionnaire comprised seven items: (a) conspicuity of the result, (b) interpretability of the result, (c) convenience for the correlation between the original image and AI-CAD result, (d) degree of visual fatigue, (e) subjective impression to improve interpretation speed, (f) subjective impression to improve interpretation accuracy, and (g) overall preference. Readers answered each question with five-point-scale scores.

### 2.7. Statistical Analysis

Statistical analyses were conducted using IBM SPSS Statistics (version 25; IBM, Armonk, NY, USA) and R (version 3.6.3; R Foundation for Statistical Computing, Vienna, Austria). To consider the clustering effect caused by multiple evaluations of single chest radiographs by multiple readers and multiple visualization methods, we used binary logistic regression with generalized estimating equations to estimate the average sensitivity, specificity, accuracy, and detection rate of the readers [15]. The detection rate of each abnormality type was also investigated. The variability of each performance metric among the five readers was evaluated using the coefficient of variation and compared using Levene’s F-test. *p*-values < 0.05 were considered statistically significant.

## 3. Results

### 3.1. Patient Demographics and Clinical Characteristics

Among 249 chest radiographs included in the study, 162 (65.1%) show clinically significant abnormalities according to the reference standard. Pulmonary air-space opacity (54.9%) is the most common abnormality, followed by pleural effusion (18.8%) and pulmonary nodule or mass (16.0%) (Table 2).

### 3.2. Examination-Level Classification Performances

Sensitivities, specificities, and accuracies of the interpretation by readers using AI-CAD with different visualization methods are described in Table 3. The highest sensitivity is observed in the combined method (71.5%; 95% CI, 65.4–76.8%), which is significantly higher than that in the closed-line method (68.2%; 95% CI, 62.2–73.6%; *p* = 0.007). Sensitivity in the heat map method does not significantly differ from that in the other two methods (70.3%; 95% CI, 64.3–75.7%; *p* = 0.383 [vs. combined method], *p* = 0.129 [vs. closed-line method]). The specificities of the interpretations do not significantly differ across the visualization methods. The accuracy of interpretation is highest in the combined method (77.0%; 95% CI, 72.6–80.9%), which is significantly higher than that in the closed-line method (75.2%; 95% CI, 70.7–79.2%; *p* = 0.037). Accuracy in the heat map method does not significantly differ from that in the other two methods (76.5%; 95% CI, 72.0–80.5%) (Table 3).

The performance of the stand-alone AI-CAD is also described in Table 3. The sensitivity (84.6%; 95% CI, 78.1–89.8%) of the stand-alone AI-CAD is significantly higher than the interpretation by readers for all visualization methods (all *p* < 0.001). Meanwhile, specificity (70.1%; 95% CI, 59.4–79.5%) is significantly lower than the interpretation by readers for all visualization methods (all *p* < 0.05). The accuracy of the stand-alone AI-CAD (77.8%; 95% CI, 70.8–83.4%) does not significantly differ from that of readers, regardless of the visualization methods.

Figure 3 and Appendix A show the accuracy of the interpretations of individual readers.

### 3.3. Lesion-Level Detection Performances

For identification of all types of abnormalities, detection rates of readers using the heat map method (66.8%; 95% CI, 61.0–72.1%) and combined method (67.5%; 95% CI, 61.7–72.8%) are significantly higher than those using the closed-line method (63.9%; 58.1–69.3%; *p* = 0.043 [vs. heat map method], *p* = 0.004 [vs. combined method]) (Table 4). The detection rates for the different types of abnormalities are described in Table 4.

The stand-alone AI-CAD exhibits a detection rate of 81.4% (95% CI, 74.8–86.8%), which is significantly higher than that of readers for all visualization methods (all *p* < 0.05) (Table 4). The detection rates of stand-alone AI-CAD for different types of abnormalities are described in Table 4.

Appendix A shows the detection rates of individual readers.

### 3.4. Variation of Performances across Readers

The coefficients of variation for sensitivity, specificity, and detection rates across the five readers are described in Table 5. The sensitivity, specificity, and detection rate show the highest degree of variation in the closed-line method (sensitivity, 0.162; specificity, 0.070; accuracy, 0.087; detection rate, 0.171) and the lowest degree of variation in the combined method (sensitivity, 0.116; specificity, 0.060; accuracy, 0.055; detection rate, 0.133). However, statistical evidence of these differences is not observed.

### 3.5. Preference Survey

Table 6 shows the results of the preference survey for three visualization methods. The rating for the overall preference is highest in the combined method. Three of five readers most preferred the combined method, while one reader preferred the closed-line method and the other preferred the heat map method. Regarding each survey question, the combined method receives the highest rating for conspicuity, interpretability, and subjective impression to improve interpretation accuracy. Meanwhile, the score for visual fatigue is also highest in the combined method.

Appendix A shows the correlation between the subjective overall preference and impression of improved accuracy versus the accuracy of interpretations.

## 4. Discussion

To enhance the performance of interpreting radiologists, the appropriate delivery and visualization of results are key components of AI-CAD. However, the optimal method for visualizing the results of AI CAD analyses has rarely been investigated. In the present study, we evaluated the performance of trainee radiologists for the identification of abnormalities in chest radiographs using AI-CAD with three different methods of visualization: (a) closed-lines along the boundary of the abnormality, (b) color-coded heat maps overlaid on the abnormality, and (c) a combination of closed-lines and heat maps. The average examination-level sensitivities are 68.2%, 70.3%, and 71.5% for the closed-line, heat map, and combined methods, respectively. A statistically significant difference is observed between the closed-line and combined methods. Meanwhile, the average specificities are similar among the three methods (89.0–89.4%).

Limited explainability is an important drawback of deep-learning-based AI algorithms, as the difficulty in understanding the logical background and factors associated with the output from the algorithm may hinder its reliability, especially for AI in healthcare [16,17,18]. AI algorithms for the detection of specific objects or findings in medical images address this explainability problem relatively simply by highlighting the location of the detected object [19,20,21]. In this regard, most currently used AIs in the field of medical imaging are designed to identify specific findings in medical images to assist physicians in practice [1,2,22]. However, for those AI-CAD applications, an appropriate explanation of results by AI-CAD and its delivery to physicians is still important because it may influence the interaction between the AI-CAD and physicians, and the performance of the physicians using the AI-CAD. In our study, the stand-alone AI-CAD exhibits higher examination-level sensitivity and lesion-level detection rate than readers using AI-CAD, indicating that a substantial proportion of true-positive detections by AI-CAD were rejected by the readers. The results suggest that improving the reliability of readers might be as important as improving the performance of AI-CAD to enhance the accuracy of interpretation by readers, which is the primary goal of AI-CAD [16,23].

The two representative methods for the visualization of AI-CAD results, the closed-line method and the heat map method, present certain advantages and disadvantages. The most important advantage of the closed-line method is the feasibility of its application in a gray-scale monitoring system. Displaying detection results without color-coded weights can be both a strength and a weakness. Although it cannot provide intuitive information for the confidence of the prediction by AI-CAD, it may help readers avoid neglecting AI-CAD detection results with low confidence. Reviewing detection with low confidence is important because the confidence of AI-CAD detection does not appropriately reflect the probability of the presence of an abnormality [13]. Meanwhile, the heat map method can help readers focus quickly on the abnormality by AI-CAD and can provide intuitive information regarding the confidence of the detection. However, detection with low confidence using AI-CAD may be neglected by the reader in the heat map method. In our study, the examination-level sensitivity of trainee radiologists was slightly higher in the heat map method than in the closed-line method, although no statistically significant difference was found (70.3% vs. 68.2%; *p* = 0.129). Lesion-level detection rates of individual abnormal findings are significantly higher using the heat map method (66.8% vs. 63.9%; *p* = 0.043). Increased attention to the color-coded heat map may have contributed to the better sensitivity of the readers.

A simple combination of the two visualization methods may have a synergistic effect in terms of the performance of readers, since it can embody the strength of both methods, that is, the increased attraction of readers for AI-CAD detection with both high and low confidence. In our study, both the examination-level sensitivity and lesion-level detection rates are significantly higher in the combined method than in the closed-line method (examination-level sensitivity, 71.5% vs. 68.2% [*p* = 0.007]; 67.5% vs. 63.9% [*p* = 0.004]). Compared to the heat map method, the combined method leads to slightly higher examination-level sensitivity and lesion-level detection rates, but no statistical evidence of a difference is observed.

Reducing inter-reader variability in interpretation accuracy is another important goal of AI-CAD [24]. In our study, although statistical evidence of differences is not observed due to the limited statistical power, the degrees of inter-reader variability of examination-level sensitivity, specificity, accuracy, and lesion-level detection rate are the lowest in the combined method.

The subjective preference of the user might be another important factor for selecting the visualization method, even though the preference or subjective impression does not perfectly correlate with the actual effectiveness (Appendix A). In the survey of the readers, the rating for overall preference is the highest in the combined method. However, the rating for visual fatigue is also the highest for the combined method. Repeated exposure to excessive information in a single overlay image (color-coded heat map, close line for the boundary, and confidence scores in numbers) may lead to fatigue in readers, especially in practice, in which a radiologist should interpret many radiographs in a limited interpretation session.

The present study has several limitations. First, our study was conducted using chest radiographs from a single institution, and only five trainee radiologists participated in the study. Therefore, the generalizability of the results is uncertain. Second, because our study was a retrospective experimental reader test, the reproducibility of our results in an actual practice situation cannot be guaranteed. Third, only a limited number of radiologists (five trainee radiologists) participated in the study, which limits the generalization of the result. Future studies with a larger number of participating radiologists might be required. Finally, the statistical power of the study might be limited because the numbers of chest radiographs and readers were relatively small.

## 5. Conclusions

In conclusion, the method of visualizing the results of AI-CAD influences the performance of radiologists’ sensitivity in identifying significant abnormalities on chest radiographs from patients with acute respiratory symptoms. The combination of the closed-line and heat map methods led to the highest examination-level sensitivity and lesion-level detection rate. A prospective study in an actual practice situation might be required to confirm the optimum method for visualizing AI-CAD results.

## Data Availability

The dataset generated during the current study is available from the corresponding author upon reasonable request.

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
