# Peer review of "Methods of Visualizing the Results of an Artificial-Intelligence-Based Computer-Aided Detection System for Chest Radiographs: Effect on the Diagnostic Performance of Radiologists"

_diagnostics, 2023, doi:10.3390/diagnostics13061089_

Round 1
Reviewer 1 Report
The manuscript has utilized four different (AI-CAD, Close line, Heat map and combined) methods to analyze the abnormalities using chest radiographs. However, there are some suggestions that would further improve the quality of the manuscript.
1. The author has provided the patient demographics and clinical characteristics of the data used in the result session. I would recommend that it be given in Section 2.1, "Patients," under "Materials and Methods.
2. Figure 4.5 should have a label on the Y-axis to assist readers in following the manuscript.
3. The author has provided the limitation and conclusion parts at the end of Section 4 Discussion. I would suggest the author put a separate heading for the conclusion and replace all the information under the conclusion heading.
4. The number of radiologists that were used in this study was only five, which is too small. I would recommend that the number of radiologists be increased in the future to more than 20, so that the results are more acceptable and realistic.
5. There are a few typos and grammatical errors. The author needs to proofread the whole manuscript carefully.
Author Response
Thank you for your review and comments for the manuscript. We made our best effort to follow the suggestions. We hope that our response and revision can alleviate the concerns. Below are post-by-point responses to your comments:
The manuscript has utilized four different (AI-CAD, Close line, Heat map and combined) methods to analyze the abnormalities using chest radiographs. However, there are some suggestions that would further improve the quality of the manuscript.
1. The author has provided the patient demographics and clinical characteristics of the data used in the result session. I would recommend that it be given in Section 2.1, "Patients," under "Materials and Methods.
→ Reflecting your suggestion, we moved the demographic information of patients to the materials and methods section.
2. Figure 4.5 should have a label on the Y-axis to assist readers in following the manuscript.
→ We modified the figure according to your suggestion.
3. The author has provided the limitation and conclusion parts at the end of Section 4 Discussion. I would suggest the author put a separate heading for the conclusion and replace all the information under the conclusion heading.
→ According to your comment, we describe the conclusion of the manuscript under a separate heading.
4. The number of radiologists that were used in this study was only five, which is too small. I would recommend that the number of radiologists be increased in the future to more than 20, so that the results are more acceptable and realistic.
→ We agree with your comment that the small number of radiologists who participated in the reader test is an important limitation of our study. We described this issue in the limitation paragraph of the discussion section. A future study with a larger number of radiologists might be required to confirm the generalizability of the result.
5. There are a few typos and grammatical errors. The author needs to proofread the whole manuscript carefully.
→ We proofread the whole manuscript and tried to fix typos and grammatical errors.
Reviewer 2 Report
1. The technical contributions of this paper are minimal, it is more like a test report of a commercialized AI-CAD system.
2. Sensitivity and specificity were used to evaluate examination-level classification performance, but the metrics such as accuracy, AUC are commonly used.
3. The heat maps are generated by the AI-CAD system for diagnosis? The corresponding classification performance of AI-CAD system should be reported.
Author Response
Thank you for your review and comments for the manuscript. We made our best effort to follow the suggestions. We hope that our response and revision can alleviate the concerns. Below are post-by-point responses to your comments:
- The technical contributions of this paper are minimal, it is more like a test report of a commercialized AI-CAD system.
→ We agree with your comment. Our study focus on the interaction between the artificial intelligence (AI)-based computer-aided detection system and human readers, rather than technical aspect of AI. However, we still believe our study has clinical implication. Currently, AI tools are used as an assistance tool for physicians, rather than a stand-alone tool. Therefore, the primary aim of an AI tool is improvement of the performance of systems. So, the interaction between AI and physician (influenced by the method of delivering the AI analysis results) is as important as the intrinsic performance of the AI tool. To date, most studies investigating AI for medical image analysis have focused on the development of AIs with nice performance and the validation of the AI performance. Meanwhile, the method of delivering the result generated by an AI system to physicians has been rarely investigated. In our study, the method of visualizing the AI result had significant influence on the performance of human readers, suggesting its relevance.
- Sensitivity and specificity were used to evaluate examination-level classification performance, but the metrics such as accuracy, AUC are commonly used.
→ Reflecting your coment, we additionally described the accuracies of interpretations. Meanwhile, since we evaluated the performance of radiologists based on the binary classification, AUC may not be an appropriate method for evaluation of performance.
- The heat maps are generated by the AI-CAD system for diagnosis? The corresponding classification performance of AI-CAD system should be reported.
→ We described the performance of stand-alone AI-CAD (Table 2), which exhibited higher sensitivity and lower specificity compared to the interpretation by radiologists. Since we used a single AI-CAD with different visualization methods, the performance of stand-alone AI-CAD was identical regardless of visualization methods (closed-line, heat map, and combined methods)
Reviewer 3 Report
Authors have modified the paper and hence it can be accepted in its current form
Author Response
Thank you for your review and positive comments.
Reviewer 4 Report
It is seen that the authors made the necessary corrections.
It is a successful study and contributes to science. It is appropriate to be published in the journal.
Author Response

(The authors gave the same response as above.)
